# Estimation of Fv/Fm in Spring Wheat Using UAV-Based Multispectral and RGB Imagery with Multiple Machine Learning Methods

**Qiang Wu [1], Yongping Zhang [1],\*, Min Xie [1],\*, Zhiwei Zhao [1], Lei Yang [2], Jie Liu [3] and Dingyi Hou [4]**

[1] College of Agronomy, Inner Mongolia Agricultural University, Huhhot 010019, China
[2] Bayannaoer Academy of Agricultural and Animal Sciences, Bayannaoer 015000, China
[3] School of Biological Science and Technology, Baotou Teachers' College, Baotou 014031, China
[4] Chifeng Forest and Grassland Protection and Development Center, Chifeng 024005, China
**\*** Correspondence: imauzyp@imau.edu.cn (Y.Z.); xiemin@imau.edu.cn (M.X.)

**Abstract:** The maximum quantum efficiency of photosystem II (Fv/Fm) is a widely used indicator of photosynthetic health in plants. Remote sensing of Fv/Fm using MS (multispectral) and RGB imagery has the potential to enable high-throughput screening of plant health in agricultural and ecological applications. This study aimed to estimate Fv/Fm in spring wheat at an experimental base in Hanghou County, Inner Mongolia, from 2020 to 2021. RGB and MS images were obtained at the wheat flowering stage using a Da-Jiang Phantom 4 multispectral drone. A total of 51 vegetation indices were constructed, and the measured Fv/Fm of wheat on the ground was obtained simultaneously using a Handy PEA plant efficiency analyzer. The performance of 26 machine learning algorithms for estimating Fv/Fm using RGB and multispectral imagery was compared. The findings revealed that a majority of the multispectral vegetation indices and approximately half of the RGB vegetation indices demonstrated a strong correlation with Fv/Fm, as evidenced by an absolute correlation coefficient greater than 0.75. The Gradient Boosting Regressor (GBR) was the optimal estimation model for RGB, with the important features being RGBVI and ExR. The Huber model was the optimal estimation model for MS, with the important feature being MSAVI2. The Automatic Relevance Determination (ARD) was the optimal estimation model for the combination (RGB + MS), with the important features being SIPI, ExR, and VEG. The highest accuracy was achieved using the ARD model for estimating Fv/Fm with RGB + MS vegetation indices on the test sets (Test set MAE = 0.019, MSE = 0.001, RMSE = 0.024, $R^2$ = 0.925, RMSLE = 0.014, MAPE = 0.026). The combined analysis suggests that extracting vegetation indices (SIPI, ExR, and VEG) from RGB and MS remote images by UAV as input variables of the model and using the ARD model can significantly improve the accuracy of Fv/Fm estimation at flowering stage. This approach provides new technical support for rapid and accurate monitoring of Fv/Fm in spring wheat in the Hetao Irrigation District.

**Keywords:** Fv/Fm; UAV; remote sensing; wheat; machine learning

## 1. Introduction

Spring wheat is an important crop in northern China, with a wide planting range and high yields. The Hetao Irrigation District is one of the major production areas for spring wheat in China, and it plays a significant role in ensuring food security [1]. Chlorophyll fluorescence has proven to be a useful indicator of photosynthetic system health and is widely employed in assessing photosynthesis [2]. Fv/Fm, a commonly used chlorophyll fluorescence parameter, represents the ratio of variable fluorescence (Fv) to maximum fluorescence (Fm) of chlorophyll. Fv represents the fluorescence emitted by open PSII reaction centers, whereas Fm represents the maximum fluorescence emitted by fully open PSII reaction centers under saturating light conditions. Fv/Fm reflects the efficiency of energy transfer within the PSII antenna and the proportion of open PSII reaction centers [3,4]. The

Fv/Fm parameter is critical in understanding the physiological status of plants and is a measure of a plant's capacity to convert light energy into chemical energy, which can offer insights into the health and productivity of crops [5]. In the case of spring wheat, Fv/Fm can be utilized to monitor crop growth and identify potential stress factors, such as water and nutrient deficiencies, that may adversely impact crop yield [6]. Traditionally, Fv/Fm has been estimated using portable instruments such as pulse amplitude modulated (PAM) fluorometers, which necessitate darkening the leaf with a leaf clamp for 15–20 min before measurement. PAM measurements can be laborious and time-consuming, particularly when large areas require monitoring. To overcome these limitations, researchers have explored the use of remote sensing technology.

Remote sensing is a powerful tool for monitoring plant growth. Remote sensing approaches retrieve chlorophyll fluorescence, which is excited by the absorption of sunlight, using spectral reflectance. Since the fluorescence emission spectrum is superimposed on leaf or canopy reflectance that can be obtained by handheld, ground-mounted, aerial, or space-borne sensors, remote sensing technique opens a new way for upscaling chlorophyll fluorescence from leaf to landscape levels. Fv/Fm has been estimated in many studies. Zhao et al. [7] collected spectral data and Fv/Fm values from potato leaves using a hyperspectral imaging system and a closed chlorophyll fluorescence imaging system, decomposed the spectral data by continuous wavelet transform (CWT), and developed an estimation model using partial least squares. Yi et al. [8] used hyperspectral and PAM fluorescence data along with correlation and regression analyses to develop Fv/Fm estimation models for aspen and cherry leaves. Jia et al. [9] calculated vegetation index using hyperspectral data to estimate Fv/Fm for wheat through linear regression. With these Fv/Fm estimation studies, the time-consuming issue of traditional fluorescence determination methods were addressed, but ground-based hyperspectral data collection was not only expensive but also incapable of estimating Fv/Fm at high spatial and temporal resolution. Unmanned aerial vehicle (UAVs) equipped with RGB sensors and multispectral sensors may solve the problem at low cost [10]. Most research on estimating Fv/Fm has primarily relied on ground-based hyperspectral measurements, with few studies employing unmanned aerial vehicles equipped with multispectral and visible sensors. This study fills a critical gap in the literature by demonstrating the feasibility of using UAV-based remote sensing to estimate Fv/Fm, offering new opportunities for efficient and high-throughput monitoring of plant health and productivity.

Machine learning techniques have revolutionized the field of data analysis by identifying complex patterns and trends that are often challenging to detect using traditional methods. In recent years, the application of machine learning methods to analyze data acquired by UAVs has gained significant traction [11–15]. However, the majority of previous studies have focused on using a limited number of machine learning methods (one to four) to estimate the desired parameters, with few investigations comparing the performance of more than twenty different techniques. Given the subtle variation of Fv/Fm and the limited spectral resolution of multispectral and RGB sensors compared to hyperspectral, it is crucial to employ multiple machine learning methods to achieve higher accuracy. This approach allows for the exploitation of the complementarity of various algorithms and enables robust and comprehensive estimation of Fv/Fm from remote sensing data. In this study, the goal is to estimate Fv/Fm in spring wheat using UAV-acquired RGB and MS remote sensing data by multiple machine learning methods to improve accuracy, which is important for rapid and accurate detection of wheat stress and timely adjustment of field management measures.

## 2. Materials and Methods

### 2.1. Study Site and Experimental Design

During the wheat flowering stage, both RGB and MS remote sensing images were obtained, resulting in the calculation of 51 vegetation indices (comprising 25 RGB and 26 multispectral). Following this, critical spectral features were extracted, while multi-

collinearity was eliminated, and feature selection was conducted to estimate the Fv/Fm values. An array of 26 machine learning techniques were utilized, with their respective performances assessed based on accuracy, stability, and interpretability. In conclusion, a high-precision UAV remote sensing monitoring model for the Fv/Fm of spring wheat in the Hetao Irrigation District was developed, thus providing a robust scientific foundation and theoretical underpinning for local agricultural advancement.

The study was conducted from 2020 to 2021 at the experimental field of the Bayannur Institute of Agriculture and Animal Husbandry Science, located in the Inner Mongolia Autonomous Region at 40°04′ N, 10°03′ E and an altitude of 1038 m above sea level (Figure 1). The soil type at the experimental site was loam, and the baseline fertility information is presented in Table 1. A split-plot design was utilized, with nitrogen (N) fertilizer application methods serving as the main plot and cultivar as the subplot. The main plot, which included five levels (CK, N1, N2, N3, and N4), featured various N application methods. N1 (0.8/0.2), N2 (0.7/0.3), N3 (0.5/0.5), and N4 (0.3/0.7) had the same N application rate of 180 kg/ha but differed in seeding fertilizer rates and follow-up fertilizer rates, while CK had no fertilizer applied. The subplot comprised three cultivars of spring wheat: "Baimai 13", "Nongmai 730", and "Nongmai 482". The experiment included a total of 15 treatments with three replications, resulting in 45 experimental plots, each measuring 12 m$^2$. The plots were arranged in randomized groups. The sowing rate was set at 300 kg/ha. Phosphorus fertilizer was applied as a basal fertilizer during sowing, and no potassium fertilizer was applied during the entire reproductive phase. Three flood irrigations were performed at the tillering, heading, and grain filling stages, each with a volume of 900 m$^3$/ha.

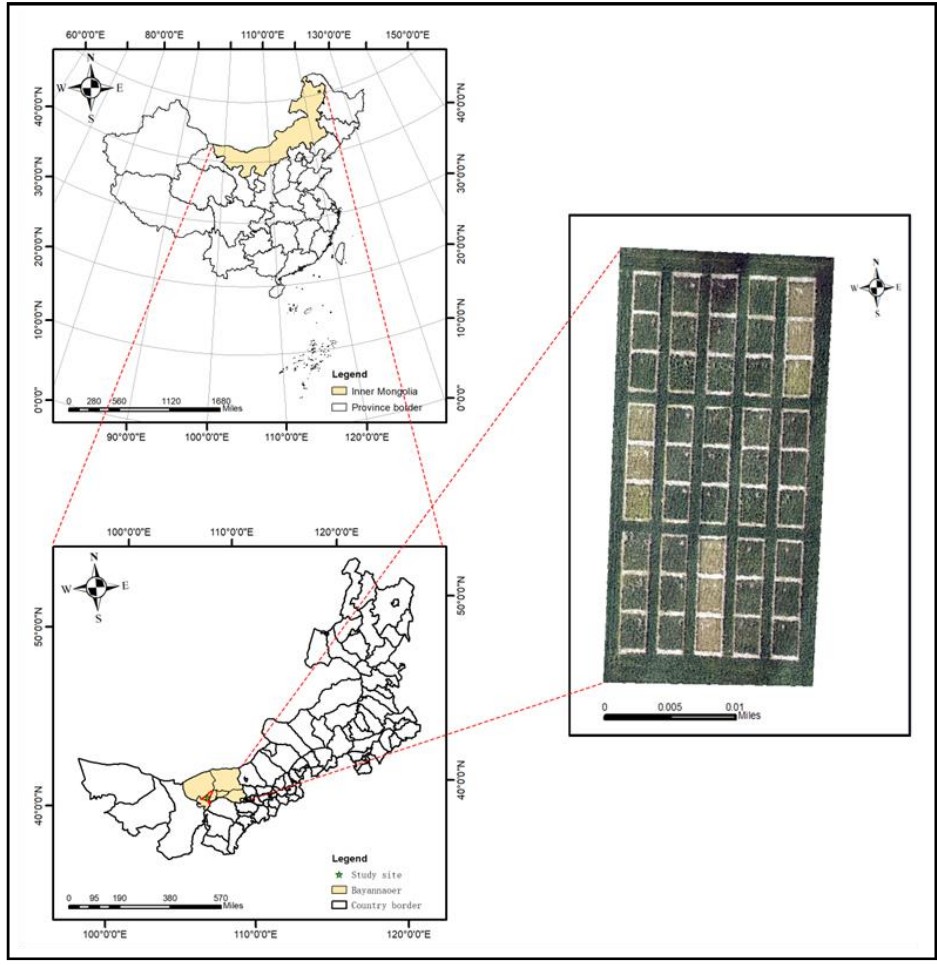

**Figure 1.** Geographical location of the experimental site.

**Table 1.** Soil base fertility information of the experimental site.

| Year | Organic Matter (g/kg) | Alkaline-N (mg/kg) | Available-P (mg/kg) | Available-K (mg/kg) | PH |
|---|---|---|---|---|---|
| 2020 | 14.31 | 59.63 | 21.32 | 117.71 | 7.62 |
| 2021 | 13.94 | 56.27 | 20.83 | 110.47 | 7.59 |

### 2.2. UAV Multispectral Data Acquisition and Processing

Remote images were obtained during the flowering stage of the wheat plant (12 June 2020; 15 June 2021) using a DJI Phantom 4 multispectral drone (Da-Jiang Innovations, Shenzhen, China). The drone (P4M, Figure 2) features 6 CMOS, including 1 color sensor (ISO: 200–800) for visible (RGB) imaging and 5 monochrome sensors (Table 2) for multispectral (MS) imaging. The images were acquired on clear and windless days, with a fixed and consistent takeoff location. The D-RTK 2 (Da-Jiang Innovations, Shenzhen, China) high-precision GNSS mobile station was utilized to assist in the positioning of the UAV and enhance its positioning accuracy. Prior to takeoff, the UAV was manually placed directly above three reflectivity gray plates of 20%, 40%, and 60%, and reflectivity plate photos were taken. The flight path was automatically planned by DJI GS Pro (Da-Jiang Innovations, Shenzhen, China) after calculating the current solar azimuth, with a flight altitude of 30 m, the ground sampling distance was 1.59 cm/pixel, a heading overlap of 85%, and a collateral overlap of 80%. Following the flight, DJI Terra (Da-Jiang Innovations, Shenzhen, China) was used to perform radiometric correction for multispectral images, followed by image stitching to generate a single-band reflectivity orthophoto. RGB images were stitched to produce color ortho images without a radiation correction step.

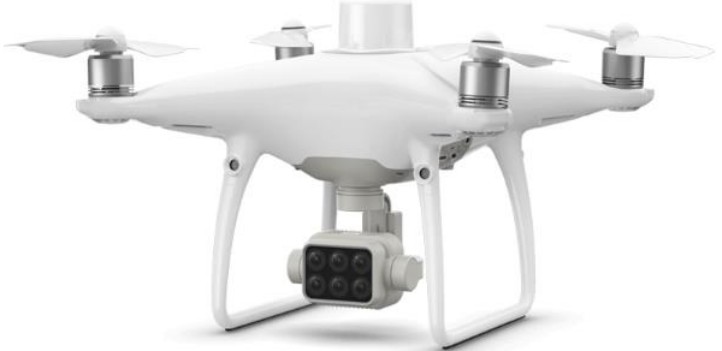

**Figure 2.** A photo for Phantom 4 multispectral.

**Table 2.** Spectral parameters of multispectral sensors.

| Band | Center Wavelength/nm | Bandwidth/nm |
|---|---|---|
| Blue (B) | 450 | 16 |
| Green (G) | 560 | 16 |
| Red (R) | 650 | 16 |
| Red Edge (RE) | 730 | 16 |
| Infrared (NIR) | 840 | 26 |

### 2.3. Construction and Selection of Spectral Indices

The digital number values for each RGB band and the reflectance of each MS band in each treatment plot were extracted using the zonal statistics function of ENVI. Subsequently, two types of vegetation indices (VIs) were computed, as presented in Tables 3 and 4.

**Table 3.** Vegetation indices based on RGB digital number value and calculation method.

| Index Name | Calculation Formula | References |
|---|---|---|
| Red (R), Green (G), Blue (B) | Raw digital number value of each band | / |
| Normalized Red | $r = R/(R + G + B)$ | / |
| Normalized Green | $g = G/(R + G + B)$ | / |
| Normalized Blue | $b = B/(R + G + B)$ | / |
| Green Red Ratio Index | $GRRI = G/R$ | / |
| Green Blue Ratio Index | $GBRI = G/B$ | / |
| Red Blue Ratio Index | $RBRI = R/B$ | / |
| Excess Red Vegetation Index | $ExR = 1.4 \times r - g$ | [16] |
| Excess Green Vegetation Index | $ExG = 2 \times g - r - b$ | [16] |
| Excess Blue Vegetation Index | $ExB = 1.4 \times b - g$ | [16] |
| Excess Green Minus Excess Red Index | $ExGR = ExG - ExR$ | [16] |
| Woebbecke Index | $WI = (G - B)/(G + R)$ | [17] |
| Normalized Difference Index | $NDI = (r - g)/(r + g + 0.01)$ | [17] |
| Color Intensity | $INT = (R + B + G)/3$ | [18] |
| Green Leaf Index 1 | $GLI1 = (2 \times G - R - B)/(2 \times G + R + B)$ | [19] |
| Green Leaf Index 2 | $GLI2 = (2 \times G - R + B)/(2 \times G + R + B)$ | [19] |
| Vegetative Index | $VEG = G/\left(R^{(2/3)} \times b^{(1/3)}\right)$ | [20] |
| Combination | $COM = 0.25 \times ExG + 0.3 \times ExGR + 0.33 \times CIVE + 0.12 \times VEG$ | [21] |
| Color Index of Vegetation | $CIVE = 0.441 \times r - 0.811 \times g + 0.3856 \times b + 18.79$ | [22] |
| Normalized Green–Red Vegetation Index | $NGRVI = (G - R)/(G + R)$ | [23] |
| Kawashima Index | $IKAW = (R - B)/(R + B)$ | [24] |
| Visible-band difference vegetation Index | $VDVI = (2 \times g - r - b)/(2 \times g + r + b)$ | [25] |
| Visible Atmospherically Resistance Index | $VARI = (g - r)/(g + r - b)$ | [26] |
| Principal Component Analysis Index | $IPCA = 0.994 \times |R - B| + 0.961 \times |G - B| + 0.914 \times |G - R|$ | [27] |
| Modified Green Red Vegetation Index | $MGRVI = (G^2 - R^2)/(G^2 + R^2)$ | [28] |
| Red Green Blue Vegetation Index | $RGBVI = (G^2 - B \times R)/(G^2 + B \times R)$ | [28] |

**Table 4.** Vegetation indices based on MS sing-band reflectance and calculation method.

| Index Name | Calculation Formula | References |
|---|---|---|
| Difference vegetation index | $DVI = R_{nir} - R_{red}$ | [29] |
| Enhanced Vegetation Index | $EVI = 2.5 \times (R_{nir} - R_{red})/(R_{nir} + 6 \times R_{red} - 7.5 \times R_{blue} + 1)$ | [29] |
| Leaf chlorophyll index | $LCI = (R_{nir} - R_{rededge})/(R_{nir} + R_{red})$ | [29] |
| Green Normalized Difference Vegetation | $GNDVI = (R_{nir} - R_{green})/(R_{nir} + R_{green})$ | [30] |
| Ratio Between NIR and Green Bands | $VI_{(nir/green)} = R_{nir}/R_{green}$ | [31] |
| Ratio Between NIR and Red Bands | $VI_{(nir/red)} = R_{nir}/R_{red}$ | [32] |
| Ratio Between NIR and Red Edge Bands | $VI_{(nir/rededge)} = R_{nir}/R_{rededge}$ | [33] |
| Napierian Logarithm of The Red Edge | $ln_{RE} = 100 \times (ln_{nir} - ln_{red})$ | [34] |
| Modified Soil-Adjusted Vegetation Index 1 | $MSAVI1 = (1 + L)\left(\frac{R_{nir} - R_{red}}{R_{nir} + R_{red} + L}\right)(L = 0.1)$ | [35] |
| Modified Soil-Adjusted Vegetation Index 2 | $MSAVI2 = R_{nir} + 0.5 - \sqrt{(2 \times R_{nir} + 1)^2 - 8 \times (R_{nir} - R_{red})}/2$ | [35] |
| Optimized Soil-Adjusted Vegetation Index | $OSAVI = (1 + 0.16) \times \frac{(R_{nir} - R_{red})}{(R_{nir} + R_{red} + 0.16)}$ | [36] |
| Modified Triangular Vegetation Index 2 | $MTVI2 = \frac{1.5 \times [1.2 \times (R_{nir} - R_{green}) - 2.5 \times (R_{red} - R_{green})]}{\sqrt{(2 \times R_{nir} + 1)^2 - (6 \times R_{nir} - 5 \times \sqrt{R_{red}}) - 0.5}}$ | [37] |
| Normalized Difference Red Edge Index | $NDRE = \frac{(R_{nir} - R_{rededge})}{(R_{nir} + R_{rededge})}$ | [38] |
| Normalized Difference Vegetation Index | $NDVI = \frac{(R_{nir} - R_{red})}{(R_{nir} + R_{red})}$ | [39] |

**Table 4.** *Cont.*

| Index Name | Calculation Formula | References |
|---|---|---|
| Modified Simple Radio | $MSR = (R_{nir} - R_{red} - 1)/\left(\sqrt{R_{nir} + R_{red}} + 1\right)$ | [40] |
| Soil-Adjusted Vegetation Index | $SAVI = \frac{(R_{nir} - R_{red})}{(R_{nir} + R_{red} + 0.5)} \times (1 + 0.5)$ | [41] |
| Simplified Canopy Chlorophyll Content Index | $SCCCI = \frac{NDRE}{NDVI}$ | [42] |
| Modified Chlorophyll Absorption Reflectance Index | $MCARI = \left(R_{rededge} - R_{red} - 0.2 \times \left(R_{rededge} - R_{green}\right)\right) \times \left(\frac{R_{rededge}}{R_{red}}\right)$ | [43] |
| Structure-Insensitive Pigment Index | $SIPI = \frac{(R_{nir} - R_{blue})}{(R_{nir} + R_{red})}$ | [44] |
| Transformed Chlorophyll Absorption Reflectance Index | $TCARI = 3 \times \left(\left(R_{rededge} - R_{red}\right) - 0.2 \times \left(R_{rededge} - R_{green}\right) \times \left(\frac{R_{rededge}}{R_{red}}\right)\right)$ | [45] |
| Normalized Difference Index | $NDI = \frac{(R_{nir} - R_{rededge})}{(R_{nir} + R_{red})}$ | [46] |
| Red-Edge Chlorophyll Index 1 | $Cl1 = \frac{R_{nir}}{R_{rededge}} - 1$ | [47] |
| Red-Edge Chlorophyll Index 2 | $Cl2 = \frac{R_{rededge}}{R_{green}} - 1$ | [48] |
| Modified Chlorophyll Absorption Reflectance Index 2 | $MCARI2 = 1.5 \times \frac{\left(2.5 \times \left(R_{nir} - R_{rededge}\right) - 1.3 \times \left(R_{nir} - R_{green}\right)\right)}{\left(2 \times (R_{nir} + 1)^2 - \left(6 \times R_{nir} - 5 \times (R_{red})^2\right) - 0.5\right)}$ | [49] |
| TCARI/OSAVI | $\frac{TCARI}{OSAVI}$ | [49] |
| MCARI/OSAVI | $\frac{MCARI}{OSAVI}$ | [49] |

## *2.4. Fluorescence Data Acquisition and Processing*

The collection of fluorescence data was synchronized with the UAV flight, and the fluorescence information of wheat canopy leaves was obtained using the Handy PEA plant efficiency analyzer (Hansatech Instruments Ltd., Norfolk, UK). In each plot, 20 leaves were randomly collected, and the average value was adopted as a representative value of the plant. Prior to collection, the target leaves were subjected to a dark treatment for 20 min using the leaf clips that were provided with the instrument.

## *2.5. Construction of Regression Model*

In this study, 26 machine learning regression models (listed in Table 5) were developed using PyCaret to estimate Fv/Fm. PyCaret is a user-friendly, open source, low-code machine learning library in Python that enables users to easily prepare data, train and evaluate machine learning models, and deploy models to production. PyCaret offers various features for data preparation, feature engineering, model training and evaluation, model interpretation, and model deployment. Additionally, it includes built-in visualizations and interactive plots that help users to interpret model results.

**Table 5.** Machine learning models.

| Model | Abbreviation | References |
|---|---|---|
| AdaBoost Regressor | ADA | [50] |
| Automatic Relevance Determination | ARD | [51] |
| Bayesian Ridge | BR | [52] |
| CatBoost Regressor | CatBoost | [53] |
| Decision Tree Regressor | DT | [54] |
| Dummy Regressor | Dummy | [55] |
| Elastic Net | EN | [56] |
| Extra Trees Regressor | ET | [57] |
| Extreme Gradient Boosting | EGB | [58] |
| Gradient Boosting Regressor | GBR | [59] |
| Huber Regressor | Huber | [60] |
| K Neighbors Regressor | KNN | [61] |
| Kernel Ridge | KR | [62] |
| Lasso Least Angle Regression | LLAR | [63] |
| Lasso Regression | Lasso | [64] |

**Table 5.** *Cont.*

| Model | Abbreviation | References |
|---|---|---|
| Least Angle Regression | LAR | [63] |
| Light Gradient Boosting Machine | LGBM | [65] |
| Linear Regression | LR | [66] |
| Multilayer Perceptron Regressor | MLP | [67] |
| Orthogonal Matching Pursuit | OMP | [68] |
| Passive Aggressive Regressor | PAR | [69] |
| Random Forest Regressor | RF | [70] |
| Random Sample Consensus | RANSC | [71] |
| Ridge Regression | Ridge | [72] |
| Support Vector Machine Regression | SVM | [73] |
| TheilSen Regressor | TR | [74] |

To prepare the data, a normalization technique was applied to transform the data into a fixed range between 0 and 1, thereby ensuring that all features were on the same scale. Multicollinearity, which refers to high correlation between multiple features, was addressed by removing highly correlated features to ensure data stability.

Feature selection was performed to select key features and reduce noise, thereby enhancing the accuracy and efficiency of the algorithm. Once all the models were built, the model with the highest accuracy was selected based on the accuracy ranking, and hyperparameter optimization was conducted to further improve model accuracy. This study used the feature selection scheme of the embedding method, implemented by calling the SelectFromModel API in sklearn, relying on the algorithmic model of random forests.

*2.6. Segmentation of Dataset and Accuracy Evaluation*

The 90 samples were randomly partitioned into a training set and a test set in a 0.7/0.3 ratio; a K-fold cross-validation (K = 5) was employed to train and optimize the model. Seven indicators were utilized to assess the accuracy of the model in the test set:

MAE (Mean Absolute Error) is a measure of the average magnitude of the errors in a set of predictions, without considering their direction. It measures the average absolute difference between the actual and predicted values. A smaller MAE indicates a more accurate prediction;

MSE (Mean Squared Error) is a measure of the average magnitude of the errors in a set of predictions, considering both the magnitude and direction of the errors. It measures the average of the squared differences between the actual and predicted values. A smaller MSE indicates a more accurate prediction;

$R^2$ (Coefficient of Determination) is a statistical measure that represents the proportion of the variance in the dependent variable that is predictable from the independent variables. In regression analysis, $R^2$ is used to evaluate the goodness of fit of the model. Generally, it ranges from 0 to 1, with a higher value indicating a better fit. An $R^2$ of 1 indicates that the model perfectly predicts the target variable, while an $R^2$ of 0 indicates that the model does not explain any variance in the target variable;

RMSE (Root Mean Squared Error) is the square root of MSE. It is a measure of the average magnitude of the errors in a set of predictions, considering both the magnitude and direction of the errors. A smaller RMSE indicates a more accurate prediction;

RMSLE (Root Mean Squared Logarithmic Error) is similar to RMSE, but instead of taking the difference between the actual and predicted values, it takes the logarithmic difference. It is used in cases where the target variable has a skewed distribution;

MAPE (Mean Absolute Percentage Error) is a measure of the average error as a percentage of the actual values. It measures the average percentage difference between the actual and predicted values. A lower MAPE indicates a more accurate prediction;

TT (Total Time) is defined as the amount of time spent on building the machine learning model. The value of TT represents the computational cost of constructing the

model, including the time spent on training and validating the model. A smaller value of TT indicates that the model has a lower computational cost and can be built more efficiently. This can be beneficial in scenarios where the model needs to be constructed in a timely manner, or when computational resources are limited. A lower TT value also indicates that the model may be more scalable and can be trained on larger datasets without excessive computational cost.

$$\text{MAE} = \frac{1}{n}\sum_{i=1}^{n}|y_i - \hat{y}| \tag{1}$$

$$\text{MSE} = \frac{\sum_{i=1}^{n}(y_i - \hat{y}_i)^2}{n} \tag{2}$$

$$\text{RMSE} = \sqrt{\frac{\sum_{i=1}^{n}(y_i - \hat{y}_i)^2}{n}} \tag{3}$$

$$\text{R}^2 = 1 - \frac{\sum_{i=1}^{n}(y_i - \hat{y}_i)^2}{\sum_{i=1}^{n}\left(y_i - \bar{y}_i\right)^2} \tag{4}$$

$$\text{RMSLE} = \sqrt{\frac{1}{n}\sum_{i=1}^{n}(log(y_i + 1) - log(\hat{y}_i + 1))^2} \tag{5}$$

$$\text{MAPE} = \frac{100\%}{n}\sum_{i=1}^{n}\left|\frac{\hat{y}_i - y_i}{y_i}\right| \tag{6}$$

where $n$ is the number of samples, $y_i$ is the observed value, $\bar{y}_i$ is the mean of the observed value, and $\hat{y}_i$ is the predicted value.

## 3. Results

### 3.1. Basic Statistical Information of the Fv/Fm Dataset

As shown in Table 6, the basic statistics of the measured Fv/Fm, indicating a range of values with a minimum of 0.550, a maximum of 0.848, a mean of 0.773, a standard deviation of 0.081, and a coefficient of variation (CV%) of 10.4. These statistics suggest that the Fv/Fm measurements demonstrate moderate variability within the total dataset. Both the training and test sets were derived from the total dataset and displayed similar ranges of Fv/Fm values. Specifically, the training set showed a minimum of 0.551, a maximum of 0.846, a mean of 0.775, a standard deviation of 0.077, and a CV% of 9.900, while the test set demonstrated a minimum of 0.550, a maximum of 0.848, a mean of 0.768, a standard deviation of 0.090, and a CV% of 11.700. The distributions of Fv/Fm values in the training and test sets were comparable, with a slightly higher mean and lower CV for the training set than the test set. These findings indicate that the training and test sets are representative of the total dataset and can be utilized for model development and validation.

**Table 6.** Basic statistics of the measured Fv/Fm.

| Dataset | Minimum | Maximum | Mean | STDEV | CV (%) |
|---|---|---|---|---|---|
| Total Dataset | 0.550 | 0.848 | 0.773 | 0.081 | 10.4 |
| Training set | 0.551 | 0.846 | 0.775 | 0.077 | 9.900 |
| Test set | 0.550 | 0.848 | 0.768 | 0.090 | 11.700 |

### 3.2. Correlation Analysis of Fv/Fm with Multispectral and RGB Vegetation Indices

The correlation coefficients between Fv/Fm values and MS, RGB vegetation indices are shown in Table 7. The highest correlation coefficients of multispectral vegetation indices were NDVI, followed by SIPI, EVI, and MSAVI2. The highest correlation coefficients of RGB vegetation indices were RGBVI, followed by VDVI, GLI, and CIVE. Most of the multi-

spectral vegetation indices and half of RGB vegetation indices showed good correlation with Fv/Fm. Overall, the correlation between the multispectral vegetation indices and Fv/Fm was higher than that of the RGB vegetation indices.

**Table 7.** Correlation coefficients between MS, RGB vegetation indices, and Fv/Fm.

| Multispectral Vegetation Indices | Correlation Coefficient | RGB Vegetation Indices | Correlation Coefficient |
|---|---|---|---|
| DVI | 0.857 | b | −0.679 |
| EVI | 0.869 | g | 0.827 |
| NDVI | 0.899 | r | 0.283 |
| GNDVI | 0.888 | GRRI | 0.309 |
| NDRE | 0.850 | GBRI | 0.737 |
| LCI | 0.797 | RBRI | 0.502 |
| OSAVI | 0.892 | INT | −0.816 |
| VI(NIR/G) | 0.784 | GRVI | 0.319 |
| VI(NIR/R) | 0.724 | NDI | −0.324 |
| VI(NIR/RE) | 0.807 | WI | 0.769 |
| lnRE | 0.86 | IKAW | 0.501 |
| MSAVI1 | 0.895 | GLI | 0.832 |
| MSAVI2 | 0.896 | GLI2 | −0.126 |
| MTVI2 | 0.863 | VARI | −0.55 |
| MSR | 0.849 | ExR | −0.435 |
| SAVI | 0.88 | ExG | 0.827 |
| SCCCI | 0.756 | ExB | −0.743 |
| MCARI | −0.793 | ExGR | 0.829 |
| MCARI2 | 0.754 | VEG | 0.767 |
| TCARI | −0.71 | IPCA | −0.821 |
| NDI | 0.865 | CIVE | −0.831 |
| CL1 | 0.807 | COM | 0.827 |
| CL2 | 0.835 | RGBVI | 0.843 |
| SIPI | 0.898 | MGRVI | 0.324 |
| TCARI/OSAVI | −0.784 | VDVI | 0.832 |
| MCARI/OSAVI | −0.893 | | |

### 3.3. Important Features Selected after Data Pre-Processing

The process of feature selection was conducted to identify the most crucial variables that significantly influence the estimation of the chlorophyll fluorescence parameter Fv/Fm in spring wheat using UAV remote sensing. Table 8 illustrates the significant features that were selected after the application of data pre-processing techniques, including the visible light vegetation indices (RGB), multispectral vegetation indices (MS), and a combination of both (RGB + MS). The pre-processing methodology involved eliminating feature collinearity and implementing tree-based feature selection methods. The table highlights that RGBVI and ExR were the vital features for the RGB dataset, MSAVI2 was the only critical feature for the MS dataset, and SIPI, ExR, and VEG were the essential features for the RGB + MS dataset.

**Table 8.** Important features selected after data pre-processing.

| VIs Type | Important Features |
|---|---|
| RGB | RGBVI, ExR |
| MS | MSAVI2 |
| RGB + MS | SIPI, ExR, VEG |

### 3.4. Model Based on RGB VIs Development and Evaluation

Table 9 presented the results of the performance evaluation of 26 machine learning models for estimating vegetation indices using RGB vegetation indices. The models were ranked according to their $R^2$ accuracy scores, with the top performing model listed first.

Evaluation metrics such as mean absolute error (MAE), mean squared error (MSE), root mean squared error (RMSE), relative mean squared logarithmic error (RMLSE), mean absolute percentage error (MAPE), and computation time (TT) were used to assess model performance. The gradient boosting regression (GBR) model achieved the highest accuracy with an $R^2$ score of 0.800, followed closely by the random forest (RF) model with an $R^2$ score of 0.795. Most of the other models performed relatively poorly, with $R^2$ scores ranging from 0.789 to $-116.050$. Notably, the worst performing models, including lasso, elastic net, least angle regression (LLAR), dummy, support vector machine (SVM), and kernel ridge regression (KR), had negative $R^2$ scores. In addition to the $R^2$ scores, the evaluation metrics indicated that the best performing models also had the lowest MAE, MSE, RMSE, RMLSE, and MAPE scores, demonstrating the models' ability to make accurate predictions with low error rates. However, there was considerable variation in computation time among the models, with some taking significantly longer than others. In conclusion, the results suggest that GBR models are the most accurate and efficient for estimating Fv/Fm using RGB vegetation indices.

**Table 9.** Accuracy assessment of different estimation models based on RGB vegetation indices.

| Model | MAE | MSE | RMSE | $R^2$ | RMLSE | MAPE | TT (s) |
|---|---|---|---|---|---|---|---|
| GBR | 0.023 | 0.001 | 0.033 | 0.800 | 0.019 | 0.032 | 0.010 |
| RF | 0.024 | 0.001 | 0.033 | 0.795 | 0.019 | 0.032 | 0.058 |
| XGB | 0.026 | 0.001 | 0.034 | 0.789 | 0.020 | 0.036 | 0.106 |
| Catboost | 0.024 | 0.001 | 0.034 | 0.785 | 0.020 | 0.033 | 0.192 |
| ET | 0.024 | 0.001 | 0.034 | 0.782 | 0.020 | 0.033 | 0.048 |
| KNN | 0.026 | 0.001 | 0.035 | 0.771 | 0.020 | 0.035 | 0.008 |
| ADA | 0.028 | 0.001 | 0.035 | 0.767 | 0.021 | 0.038 | 0.018 |
| Huber | 0.030 | 0.002 | 0.039 | 0.711 | 0.022 | 0.040 | 0.006 |
| Ridge | 0.031 | 0.002 | 0.039 | 0.707 | 0.023 | 0.041 | 0.006 |
| LR | 0.030 | 0.002 | 0.039 | 0.707 | 0.023 | 0.040 | 0.006 |
| LAR | 0.030 | 0.002 | 0.039 | 0.707 | 0.023 | 0.040 | 0.006 |
| BR | 0.030 | 0.002 | 0.039 | 0.707 | 0.023 | 0.041 | 0.006 |
| OMP | 0.031 | 0.002 | 0.039 | 0.707 | 0.022 | 0.041 | 0.004 |
| ARD | 0.031 | 0.002 | 0.039 | 0.707 | 0.022 | 0.041 | 0.006 |
| DT | 0.031 | 0.002 | 0.040 | 0.688 | 0.023 | 0.042 | 0.004 |
| TR | 0.036 | 0.003 | 0.049 | 0.572 | 0.029 | 0.051 | 0.178 |
| PAR | 0.042 | 0.003 | 0.050 | 0.528 | 0.029 | 0.056 | 0.006 |
| LGBM | 0.049 | 0.004 | 0.063 | 0.283 | 0.036 | 0.067 | 0.018 |
| RANSC | 0.043 | 0.005 | 0.065 | 0.195 | 0.037 | 0.062 | 0.008 |
| MLP | 0.055 | 0.005 | 0.067 | 0.158 | 0.038 | 0.072 | 0.014 |
| lasso | 0.058 | 0.006 | 0.076 | $-0.059$ | 0.045 | 0.082 | 0.006 |
| EN | 0.058 | 0.006 | 0.076 | $-0.059$ | 0.045 | 0.082 | 0.006 |
| LLAR | 0.058 | 0.006 | 0.076 | $-0.059$ | 0.045 | 0.082 | 0.008 |
| Dummy | 0.058 | 0.006 | 0.076 | $-0.059$ | 0.045 | 0.082 | 0.004 |
| SVM | 0.072 | 0.006 | 0.077 | $-0.104$ | 0.044 | 0.093 | 0.006 |
| KR | 0.792 | 0.632 | 0.795 | $-116.050$ | 0.523 | 1.030 | 0.006 |

### 3.5. Model Based on MS VIs Development and Evaluation

Table 10 presented an accuracy assessment of 26 machine learning models for estimating vegetation indices using multi-spectral data. The top seven models with the highest $R^2$ scores, ranging from 0.860 to 0.849, were Huber, LR, Ridge, LAR, OMP, BR, ARD, and TR. These models had the lowest MAE, MSE, RMSE, and RMLSE values among all the models, indicating that they produced the most accurate estimates of vegetation indices. The computation time for these models ranged from 0.004 to 0.316 s, with the LR model having the longest computation time. The next group of models, with $R^2$ scores ranging from 0.794 to 0.684, included KNN, RF, CatBoost, ADA, GBR, ET, and PAR. These models had higher MAE, MSE, RMSE, and RMLSE values than the top models, indicating that they produced less accurate estimates of vegetation indices. The computation time for these

models ranged from 0.006 to 0.190 s. The last group of models, with $R^2$ scores ranging from 0.668 to −0.567, included XGB, DT, RANSC, LGBM, SVM, Lasso, EN, LLAR, Dummy, MLP, and KR. These models had the lowest $R^2$ scores and the highest MAE, MSE, RMSE, and RMLSE values, indicating that they produced the least accurate estimates of Fv/Fm. The computation time for these models ranged from 0.004 to 0.298 s, with the KR model having the longest computation time. Overall, the Huber model was the most accurate for estimating Fv/Fm using multispectral vegetation indices.

**Table 10.** Accuracy assessment of different estimation models based on MS vegetation indices.

| Model | MAE | MSE | RMSE | $R^2$ | RMLSE | MAPE | TT (s) |
|---|---|---|---|---|---|---|---|
| Huber | 0.021 | 0.001 | 0.027 | 0.860 | 0.015 | 0.028 | 0.006 |
| LR | 0.022 | 0.001 | 0.027 | 0.860 | 0.015 | 0.029 | 0.316 |
| Ridge | 0.022 | 0.001 | 0.027 | 0.860 | 0.015 | 0.029 | 0.264 |
| LAR | 0.022 | 0.001 | 0.027 | 0.860 | 0.015 | 0.029 | 0.004 |
| OMP | 0.022 | 0.001 | 0.027 | 0.860 | 0.015 | 0.029 | 0.004 |
| BR | 0.022 | 0.001 | 0.027 | 0.860 | 0.015 | 0.029 | 0.004 |
| ARD | 0.022 | 0.001 | 0.027 | 0.860 | 0.015 | 0.029 | 0.006 |
| TR | 0.022 | 0.001 | 0.028 | 0.849 | 0.016 | 0.029 | 0.020 |
| KNN | 0.026 | 0.001 | 0.033 | 0.794 | 0.019 | 0.035 | 0.006 |
| RF | 0.032 | 0.001 | 0.038 | 0.737 | 0.022 | 0.043 | 0.054 |
| Catboost | 0.033 | 0.002 | 0.039 | 0.723 | 0.022 | 0.045 | 0.190 |
| ADA | 0.031 | 0.002 | 0.039 | 0.716 | 0.023 | 0.042 | 0.012 |
| GBR | 0.034 | 0.002 | 0.040 | 0.704 | 0.023 | 0.046 | 0.010 |
| ET | 0.034 | 0.002 | 0.041 | 0.695 | 0.023 | 0.046 | 0.044 |
| PAR | 0.035 | 0.002 | 0.041 | 0.684 | 0.023 | 0.046 | 0.006 |
| XGB | 0.036 | 0.002 | 0.042 | 0.668 | 0.024 | 0.049 | 0.132 |
| DT | 0.037 | 0.002 | 0.043 | 0.652 | 0.025 | 0.050 | 0.006 |
| RANSC | 0.035 | 0.003 | 0.047 | 0.546 | 0.027 | 0.049 | 0.006 |
| LGMB | 0.045 | 0.004 | 0.062 | 0.299 | 0.036 | 0.064 | 0.016 |
| SVM | 0.064 | 0.005 | 0.069 | 0.097 | 0.039 | 0.082 | 0.006 |
| Lasso | 0.058 | 0.006 | 0.076 | −0.059 | 0.045 | 0.082 | 0.298 |
| EN | 0.058 | 0.006 | 0.076 | −0.059 | 0.045 | 0.082 | 0.006 |
| LLAR | 0.058 | 0.006 | 0.076 | −0.059 | 0.045 | 0.082 | 0.004 |
| Dummy | 0.058 | 0.006 | 0.076 | −0.059 | 0.045 | 0.082 | 0.004 |
| MLP | 0.076 | 0.008 | 0.090 | −0.567 | 0.052 | 0.101 | 0.016 |
| KR | 0.784 | 0.619 | 0.787 | −113.552 | 0.528 | 1.024 | 0.006 |

*3.6. Model Based on RGB and MS VIs Development and Evaluation*

Table 11 presented an accuracy assessment of 26 machine learning models for the estimation of vegetation indices (VIs) using both RGB and multispectral data. The ARD and OMP models achieved the highest $R^2$ accuracy scores of 0.868, followed closely by the Ridge, LR, LAR, BR, and Huber models, all with $R^2$ values of 0.858. The Tr model obtained an $R^2$ of 0.849, while RANSC, KNN, RF, ET, ADA, Catboost, and XGB models had $R^2$ values ranging from 0.830 to 0.723. The GBR and DT models had $R^2$ values of 0.721 and 0.690, respectively. The PAR model had an $R^2$ of 0.593, indicating lower accuracy than the previous models. On the other hand, the LGMB, Lasso, EN, LLAR, and Dummy models had negative $R^2$ values, indicating poor accuracy. Moreover, the SVM and MLP models had low $R^2$ values of −0.073 and −0.292, respectively. The KR model had the worst performance with high MAE, MSE, RMSE, and RMLSE values and a very low $R^2$ value of −118.159. In summary, the ARD and OMP models demonstrated the same highest accuracy for estimating Fv/Fm using both MS and RGB data. However, the computation time of OMP was found to be higher than that of ARD. Therefore, the optimal model is ARD.

**Table 11.** Accuracy assessment of different estimation models based on RGB and MS indices.

| Model | MAE | MSE | RMSE | $R^2$ | RMLSE | MAPE | TT (s) |
|---|---|---|---|---|---|---|---|
| ARD | 0.021 | 0.001 | 0.026 | 0.868 | 0.015 | 0.028 | 0.006 |
| OMP | 0.021 | 0.001 | 0.026 | 0.868 | 0.015 | 0.028 | 0.007 |
| Ridge | 0.022 | 0.001 | 0.027 | 0.858 | 0.015 | 0.029 | 0.006 |
| LR | 0.022 | 0.001 | 0.027 | 0.858 | 0.015 | 0.029 | 0.514 |
| LAR | 0.022 | 0.001 | 0.027 | 0.858 | 0.015 | 0.029 | 0.006 |
| BR | 0.022 | 0.001 | 0.027 | 0.858 | 0.015 | 0.029 | 0.006 |
| Huber | 0.022 | 0.001 | 0.027 | 0.857 | 0.016 | 0.029 | 0.006 |
| TR | 0.022 | 0.001 | 0.028 | 0.849 | 0.016 | 0.030 | 0.178 |
| RANSC | 0.024 | 0.001 | 0.030 | 0.830 | 0.017 | 0.032 | 0.010 |
| KNN | 0.023 | 0.001 | 0.030 | 0.826 | 0.017 | 0.031 | 0.006 |
| RF | 0.025 | 0.001 | 0.033 | 0.800 | 0.019 | 0.033 | 0.052 |
| ET | 0.025 | 0.001 | 0.034 | 0.785 | 0.020 | 0.033 | 0.050 |
| ADA | 0.027 | 0.001 | 0.036 | 0.756 | 0.021 | 0.037 | 0.020 |
| Catboost | 0.026 | 0.001 | 0.036 | 0.753 | 0.021 | 0.036 | 0.214 |
| XGB | 0.027 | 0.002 | 0.038 | 0.723 | 0.022 | 0.037 | 0.092 |
| GBR | 0.028 | 0.002 | 0.039 | 0.721 | 0.022 | 0.037 | 0.012 |
| DT | 0.032 | 0.002 | 0.041 | 0.690 | 0.024 | 0.043 | 0.006 |
| PAR | 0.039 | 0.002 | 0.047 | 0.593 | 0.027 | 0.051 | 0.006 |
| LGBM | 0.043 | 0.003 | 0.055 | 0.429 | 0.032 | 0.060 | 0.018 |
| Lasso | 0.058 | 0.006 | 0.076 | −0.059 | 0.045 | 0.082 | 0.006 |
| EN | 0.058 | 0.006 | 0.076 | −0.059 | 0.045 | 0.082 | 0.008 |
| LLAR | 0.058 | 0.006 | 0.076 | −0.059 | 0.045 | 0.082 | 0.006 |
| Dummy | 0.058 | 0.006 | 0.076 | −0.059 | 0.045 | 0.082 | 0.004 |
| SVM | 0.071 | 0.006 | 0.076 | −0.073 | 0.043 | 0.092 | 0.006 |
| MLP | 0.063 | 0.008 | 0.082 | −0.292 | 0.047 | 0.085 | 0.020 |
| KR | 0.799 | 0.646 | 0.803 | −118.159 | 0.512 | 1.042 | 0.006 |

*3.7. Hyperparameter Optimization of the Highest Accuracy Model in Three Modes*

Table 12 and Figure 3 show the hyperparameter-optimized performance of the most accurate machine learning models in three different image types (RGB, MS and MS + RGB). The results show that the performance of these models varies according to the different image types and data sets. Notably, the ARD model achieves the highest accuracy on the test set of MS + RGB VIs mode (test set MAE = 0.019, MSE = 0.001, RMSE = 0.024, $R^2$ = 0.925, RMSLE = 0.014 and MAPE = 0.026).

**Table 12.** Hyperparameter optimization of the highest accuracy model in three image types.

| Image Type | Model | Dataset | MAE | MSE | RMSE | $R^2$ | RMLSE | MAPE |
|---|---|---|---|---|---|---|---|---|
| RGB | GBR | Training set | 0.014 | 0.000 | 0.021 | 0.925 | 0.012 | 0.019 |
| | | Test set | 0.027 | 0.001 | 0.036 | 0.838 | 0.020 | 0.037 |
| MS | Huber | Training set | 0.021 | 0.001 | 0.027 | 0.878 | 0.015 | 0.028 |
| | | Test set | 0.018 | 0.001 | 0.025 | 0.920 | 0.014 | 0.024 |
| MS + RGB | ARD | Training set | 0.021 | 0.001 | 0.026 | 0.882 | 0.015 | 0.028 |
| | | Test set | 0.019 | 0.001 | 0.024 | 0.925 | 0.014 | 0.026 |

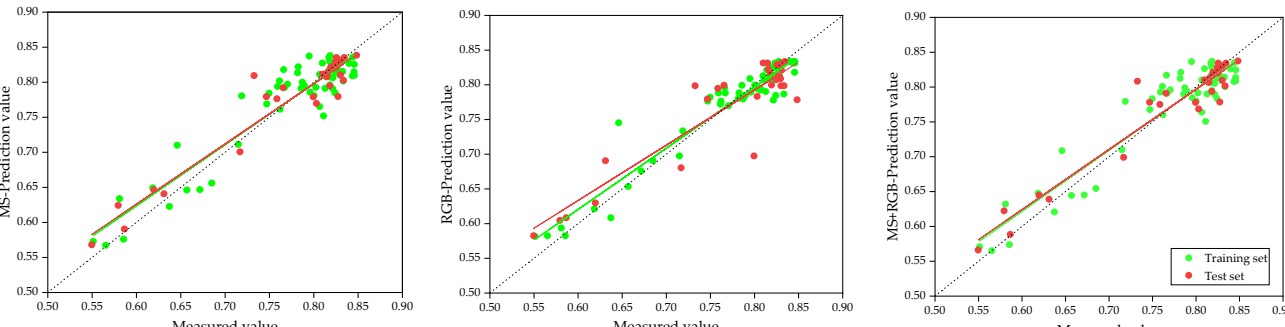

**Figure 3.** Calibration and validation results of the highest accuracy model in MS, RGB, RGB + MS data after hyperparameter optimization.

## 4. Discussion

Chlorophyll fluorescence is a potent indicator of photosynthesis, where Fv/Fm serves as a measure of the maximum photochemical efficiency of photosystem II in chloroplasts and can be utilized as an indicator of plant health [75]. For many plant species, the optimal Fv/Fm range is approximately between 0.79 and 0.84, with lower values indicating higher plant stress [76]. The wheat flowering period is a critical stage of growth and development, as it signifies the transition of wheat plants from vegetative to reproductive growth stages, where the focus of growth shifts from leaves and roots to flowers and seeds [77]. During this period, wheat plants require significant amounts of nutrients and energy to support the normal development and maturation of flowers and seeds [78]. Therefore, providing sufficient water and nutrients is crucial for wheat plants during the flowering period. Stress during this phase can significantly affect the yield and quality of wheat [79]. Monitoring Fv/Fm during the flowering period is crucial in detecting stress and implementing timely agricultural management practices to optimize wheat production and achieve high yields.

In this study, both multispectral vegetation indices and RGB vegetation indices were found to be effective in estimating Fv/Fm values. The $R^2$ of test set was 0.920 for multispectral vegetation indices and 0.838 for RGB vegetation indices, indicating that the former had a higher accuracy in estimating Fv/Fm values. Multispectral images acquired by UAVs can provide information about the spectral reflectance of vegetation, which changes simultaneously in the canopy when stress occurs [80], and this can be used to estimate Fv/Fm. The key idea of this approach is that Fv/Fm is related to the fluorescence yield of photosystem II, which in turn is related to the chlorophyll content of leaves. The chlorophyll content can be estimated from the reflectance in the red and near-infrared (NIR) bands [81]. The important spectral index MSAVI2 extracted from the multispectral estimation in this study was calculated precisely from the red and NIR bands. In addition to multispectral images, RGB images can also be used to estimate Fv/Fm values by color information. In general, the color of plants changes when they are under stress, and although some color changes are difficult for the human eye to observe, color values can be quantified by computer technology [82]. However, due to the lack of vegetation-sensitive red edge and infrared bands, the RGB images do not provide enough spectral information, so the estimated Fv/Fm is not as accurate as the MS images.

This study shows that the machine learning models have high accuracy and stability and can effectively use RGB and MS data to estimate Fv/Fm, which is consistent with the findings of other studies [83] using machine learning for remote sensing estimation. The effects of different machine learning models and different types of vegetation indices on the accuracy of Fv/Fm estimation are obvious. Firstly, different types of vegetation indices can provide different information in estimating Fv/Fm, and different machine learning models have different adaptation and fitting ability to different datasets [84]. The MS + RGB model exhibits superior accuracy compared to both the MS and RGB models in estimating Fv/Fm. This suggests that the integration of RGB and MS data provides benefits in enhancing the accuracy of the estimation. Additionally, the findings suggest that utilizing multiple

data sources can enhance the accuracy of the model as compared to relying solely on single-source data. These results are consistent with previous studies that have reported improved accuracy in multi-source data estimation [85,86]. However, the improvement in accuracy when combining (MS + RGB) is only marginally better than using MS alone in this study. The ARD model cannot be quantified for the percentage contribution of RGB and MS, so the random-forest-model-based feature importance assessment was implemented, the result was shown in Figure 4. The importance of multispectral vegetation index in the model was higher than that of RGB vegetation indices, and the RGB contributed less valid information.

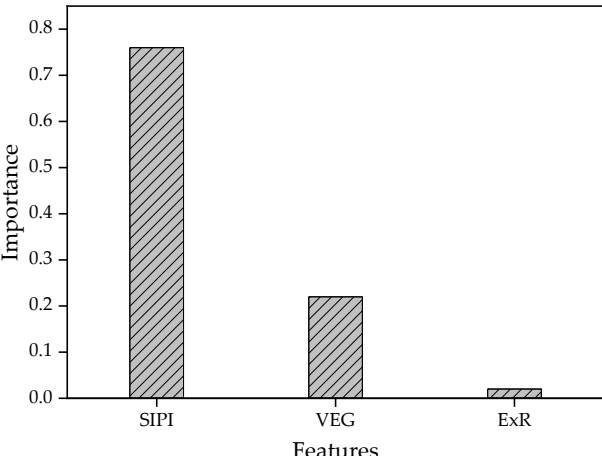

**Figure 4.** Feature importance assessment based on random forest model.

Zhao et al. [7] developed two estimation models for Fv/Fm of potato leaves with test set $R^2$ of 0.807 and 0.822 and RMSE of 0.018 and 0.017, respectively. Yi et al. [8] developed a generalized estimation model of Fv/Fm for poplar leaves and cherry leaves with $R^2 = 0.88$. In a study [9] estimating Fv/Fm in winter wheat, training set $R^2 = 0.50$, RMSE = 0.012, test set $R^2 = 0.55$, RMSE = 0.014. In these previous estimation studies of Fv/Fm based on hyperspectral, the highest $R^2$ was of 0.88, and the highest accuracy combined model without hyperparameter optimization in this study had $R^2$ of 0.868, which shows that the estimation accuracy of multispectral and RGB is not as good as hyperspectral. However, Yang et al. [87] showed that hyperparameter optimization can be helpful in improving the estimation accuracy of the model. In this study, the test set $R^2$ of both the MS estimation and the estimation of the MS and RGB combination reaches 0.92 after hyperparameter optimization, indicating that the accuracy gap caused by the sensors can be narrowed or even surpassed by algorithm optimization. Negative values of $R^2$ were observed in all three estimation models, which are usually uncommon, indicating that the prediction results using the model are worse than the estimation results using the mean, because the mean reflects the central tendency of the data, while the prediction results of the model deviate more from the true value than the mean [88]. Notably, the performance of linear regression (LR), ridge regression (Ridge), least angle regression (LAR), and orthogonal matching pursuit (OMP) models was almost identical. Moreover, the reason for this was that only MSAVI2 was screened in the multispectral vegetation indices and LR, Ridge, LAR, and OMP are all linear models, indicating that different linear models may not have a significant effect on the accuracy of estimation when a single vegetation index was used as a characteristic variable.

The present study has important practical implications for agriculture and environmental monitoring. By demonstrating the feasibility of using UAV-based remote sensing to estimate Fv/Fm in spring wheat, our research offers new opportunities for efficient and high-throughput monitoring of plant health and productivity. This can help to improve crop management strategies, enhance yield and quality of agricultural products, and reduce the environmental impact of farming practices. In addition, the integration of machine

learning methods with multispectral and RGB imagery can further enhance the accuracy and reliability of Fv/Fm estimation, enabling more precise and targeted interventions in crop management.

## 5. Conclusions

Integration of multiple machine learning methods with multispectral and RGB imagery acquired from UAV-based remote sensing can improve the accuracy and reliability of Fv/Fm estimation in spring wheat during the flowering stage. The important features and the optimal Fv/Fm estimation models for different types remote sensing images were different: with gradient boosting regressor (GBR) as the optimal estimation model for RGB, the important features were RGBVI and ExR; with Huber as the optimal estimation model for MS, the important feature was MSAVI2; and automatic relevance determination (ARD) as the optimal estimation model for combination (MS + RGB), the important features were SIPI, ExR, VEG. The highest accuracy was achieved using the ARD model for estimating Fv/Fm with RGB + MS vegetation indices (Test set MAE = 0.019, MSE = 0.001, RMSE = 0.024, $R^2$ = 0.925, RMSLE = 0.014, MAPE = 0.026). Based on the results of this study, there is great potential for the use of remote sensing and machine learning for efficient and sustainable plant health monitoring and management.

In conclusion, while the present study provides a valuable contribution to the use of remote sensing and machine learning for estimating Fv/Fm in wheat during the flowering stage, there are several limitations. Firstly, the study was conducted at a single ecological site over a two-year period, which may limit the generalizability of the findings to other regions and ecosystems. Future studies conducted at multiple sites with varying environmental conditions would provide a more comprehensive understanding of the applicability of the proposed methods. Secondly, the study focused solely on the flowering stage of wheat growth, which is only one phase of the crop's development. It is possible that the performance of the proposed remote sensing and machine learning methods may differ at other stages of wheat growth, such as germination, tillering, or grain filling. Further research is needed to examine the feasibility and accuracy of the methods across multiple growth stages. Thirdly, while the proposed methods show promise for estimating Fv/Fm in wheat, it should be noted that these methods may not be directly applicable to other plant species or crops. The optimal settings and parameters for remote sensing and machine learning may vary depending on the physiological and structural characteristics of the target plants. Future studies should explore the generalizability and adaptability of the methods to other crops and plant species.

**Author Contributions:** Conceptualization, Q.W. and Y.Z.; methodology, Q.W.; software, M.X.; validation, Z.Z. and J.L.; formal analysis, Z.Z.; investigation, D.H.; resources, L.Y.; data curation, D.H. and J.L; writing—original draft preparation, Q.W.; writing—review and editing, Y.Z.; visualization, M.X.; supervision, Y.Z.; project administration, Y.Z.; funding acquisition, Y.Z. and M.X. All authors have read and agreed to the published version of the manuscript.

**Funding:** This research was funded by Inner Mongolia "science and technology" action focus on special "Yellow River Basin durum wheat industrialization capacity enhancement" (NMKJXM202201-4); Inner Mongolia "science and technology" action focus on special "Research and Application of Key Technologies for Production and Processing of Durum Wheat and Products in Hetao irrigation area" (NMKJXM202111-3) and Inner Mongolia Natural Science Foundation of China "Research on nitrogen nutrition diagnosis of spring wheat in Hetao irrigation area based on UAV mapping technology" (2021MS03089).

**Data Availability Statement:** Not applicable.

**Acknowledgments:** We sincerely appreciate the assistance of the Wheat Research Institute of the Inner Mongolia Bayannur Academy of Agriculture and Animal Science.

**Conflicts of Interest:** The authors declare no conflict of interest.

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
