# Peer review of "Estimation of Fv/Fm in Spring Wheat Using UAV-Based Multispectral and RGB Imagery with Multiple Machine Learning Methods"

_agronomy, doi:10.3390/agronomy13041003_

Round 1

Reviewer 1 Report

The Authors present an interesting manuscript and the manuscript brings new elements to existing knowledge about performance of Estimation of Fv/Fm in Spring Wheat by UAV RGB and MS Imagery using Multiple Machine Learning Methods. The manuscript is written with clear understanding of the project addressed. It includes a well-crafted abstract and an exhaustive introduction that justifies the research undertaken. The introduction points to the deficiencies in the literature on the subject. The aim is clearly defined. Modern analytical methods were used in the research. The discussion of the results is well prepared. The conclusions are well-defined. The illustrative material is appropriate. However, my recommendation is to accept publication after the following minor revisions:

Introduction:

Based on your objectives, please compare how your study is different from those that have already been published.

Discussion

- The discussion section needs to be tightened and supported with the obtained data and that relevant published studies.

Conclusions:

 -Add main finding of this study

- Add recommendation for future studies

Reviewer 2 Report

Dear authors,

You have employed MS (multispectral) and RGB imagery for remote sensing of Fv/Fm, which helps in screening plant health in agricultural and ecological applications. The topic is interesting and relatively well presented. However, I have decided to reject your manuscript due to the following reason. You can, of course, apply my comments and improve its quality for further submissions to this or other journals:

1- The introduction should introduce the topic with a short review of related literature and address any knowledge gaps that this manuscript fills. This can be considered the novelty of the study.

2- I personally don't believe that comparing different ML methods is a novelty anymore.

3- The materials and methods section is well presented. However, the feature selection method has not been introduced. The spatial resolution of the images has not been also mentioned.

4- The main problem with this manuscript is the discussion section. It should discuss the results and compare them with other studies, while avoiding repeating the introduction. Most of the discussion, such as the first paragraph and half of the second one, is redundant and can be removed or moved to the introduction. Besides, you have not discussed why some methods showed higher performance than others. Why models developed on selected features were better than others?

5- You have mentioned that ARD can not perform the variable importance, hence there is no VIP in your study. You could use PLSR, RF, Cubist, and SVR to perform a VIP and find the important variables.

I have also some minor comments:

6- The final line of the second paragraph needs citation.

7-” However, previous studies [11–15] have largely focused on using 1-4 machine learning methods…” what is 1-4? likely refers to the use of four specific machine learning methods (perhaps numbered 1-4 in previous studies). However, this should be clarified or explained further in the text.

8- Each abbreviation should be defined once it appears in the text.

9- It is generally preferable to use decimal or percentage values rather than ratios when discussing numerical data. Thus, using "0.7/0.3" or "70/30%" instead of "7:3" would be clearer and more standard.

10- Some sentences seem to be incomplete in the fourth paragraph of the discussion „Notably, the performance of linear regression (LR), ridge regression (Ridge), least angle regression (LAR) and orthogonal matching pursuit (OMP) models were almost identical in...“ and „The reason is that only MSAVI2 was screened in the multispectral vegetation indices and LR, Ridge, LAR, and OMP are all linear models, indicating that different linear models may not have a significant effect on the accuracy of estimation when single vegetation…

Reviewer 3 Report

The experimental work presented in the Manuscript, entitled Estimation of Fv/Fm in Spring Wheat by UAV RGB and MS Imagery using Multiple Machine Learning Methods". The article reports that The results showed that the important features and optimal Fv/Fm estimation models for different remote sensing image types varied. TheGradient Boosting Regressor (GBR) was the optimal estimation model for RGB, with the important features being RGBVI and ExR. The Huber model was the optimal estimation model for MS, with the important feature being MSAVI2. The Automatic Relevance Determination (ARD) was the optimal estimation model for the combination (RGB+MS), with the important features being SIPI, ExR, and VEG. The highest accuracy was achieved using the ARD model for estimating Fv/Fm with RGB
+ MS vegetation indices on the test sets (Test set MAE = 0.019, MSE = 0.001, RMSE = 0.024, R2 = 0.925, RMSLE = 0.014, MAPE = 0.026. There are several shortcomings and modifications that should be included in order to enhance the manuscript for the readers.

Title

1-      Please modified UAV RGB to UAV based on RGB, and MS to multispectral. Also arrange the tools as you presented in results. Mean that multispectral should come before RGB.

Abstract

2-      Please do not use pronouns in the formulation of the sentence. For example, we compared.

3-       This sentence should be removed (The model was evaluated using various metrics, including MAE, MSE, RMSE, R2, RMSLE, and MAPE).

4-      This sentence should be rephrased or removed (The results showed that the important features and optimal Fv/Fm estimation models for different remote sensing image types varied). 5-      What about the relationships between the vegetation indices based on RGB and MS with Fv/Fm.

Introduction

6-      Please add the hint about the basic of remote sensing for detect Fv/Fm?

7-      These sentence are related to Materials and Methods (We acquired RGB and MS remote sensing images during wheat flowering in 2020-2021 and calculated a total of 51 vegetation indices (25 RGB and 26 multispectral). We then extracted the important spectral features, removed multicollinearity and performed feature selection to estimate Fv/Fm. We employed 26 machine learning techniques and compared their performance in terms of accuracy, stability, and interpretability. Finally, we established a high-precision UAV remote sensing monitoring model of Fv/Fm of spring wheat in the Hetao Irrigation District, providing a scientific basis and theoretical foundation for local agricultural development).

8-      Please highlight in introduction, what is the novelty (originality) of the work? And what is new in your work that makes a difference in the body of knowledge?

Materials and Methods

9-      The title of figure 1 should be added under the figure.

10-  Figure 1 should be improved.

11-  In table 3, what do you mean about (/)

Results

12-  The relationships between the vegetation indices based on RGB and MS with Fv/Fm should be presented to the performance for each vegetation index.

13-  Figure 2 should be improved.

Discussions

14-  The author presented the discussion in good way.

15-  Please, write the practical applications of your work in a separate section, before the conclusions and provide your good perspectives.

Conclusion 

115-  Please write about the limitations of this work in details in conclusion section.

Round 2

Reviewer 2 Report

Dear authors,

You have addressed most of my comments and hence I've decided to accept the manuscript after considering the following minor tips:

1- Please double-check the manuscript title. It seems grammatically incorrect.

2- Modify the first sentence of the second paragraph of the introduction. It seems grammatically incorrect.

Reviewer 3 Report

The authors improved the manuscript according to my comments.
